# Experimental Study on Mechanical Characteristics and Fracture Patterns of Unfrozen/Freezing Saturated Coal and Sandstone

**DOI:** 10.3390/ma12060992

**Published:** 2019-03-26

**Authors:** Chong Wang, Shuangyang Li, Tongwei Zhang, Zhemin You

**Affiliations:** 1School of Civil Engineering and Mechanics, Lanzhou University, Lanzhou 730000, China; wangchong@lzu.edu.cn (C.W.); ztw@lzu.edu.cn (T.Z.); 2Key Laboratory of Mechanics on Disaster and Environment in Western China, MOE, Lanzhou University, Lanzhou 730000, China; 3State Key Laboratory of Frozen Soil Engineering, Northwest Institute of Eco-Environment and Resources, Chinese Academy of Sciences, Lanzhou 730000, China; youzhemin@lzb.ac.cn

**Keywords:** mechanical properties, water-bearing coal and sandstone, temperature effect, failure mode

## Abstract

The thermomechanical behavior of coal and sandstone during excavation using the freezing method is a new challenge for coal mining and geotechnical engineering. In this paper, the influence of temperature on the mechanical characteristics and fracture patterns of two types of saturated rocks (coal and sandstone) were investigated. A series of laboratory tests, including the Brazilian tensile strength (BTS), uniaxial compressive strength (UCS), and triaxial compressive strength (TCS), were conducted at temperatures of 20, −4, −10, and −15 °C. The results indicated a significant increase in their strength when the temperature was reduced from 20 to −15 °C, especially near the phase-transition point. Then, a theoretical model was proposed to predict rock strength change with temperature, based on the phase-transition theory. To evaluate this model, the predicted results were compared with experimental data, where a good correlation was identified. In addition, four failure patterns were observed in indirect tensile tests (i.e., layer activation, central fracture, noncentral fracture, and central and layer activation), and three types of failure modes in compression tests (i.e., axial splitting, shearing along a single plane, multiple fracturing). The evolution of the rock damage was divided into four stages: Crack closure, fracture initiation, critical energy release, and rupture. These results could be applied to evaluate and predict the mechanical behavior of saturated coal and sandstone during excavation using the freezing method.

## 1. Introduction

In recent years, since coal mining was extended to deep and thick alluvium layers, the freezing method has been considered an effective way to resist the high initial stress in these circumstances [1,2,3,4]. The freezing method is an excavation technique using artificial refrigeration to temporarily consolidate the wall rock in an aquifer. The strengths (e.g., compressive and tensile strength) of rocks in frozen and unfrozen states are significantly different [5,6]. Hence, more attention should be paid to the mechanical strengths and failure modes of both frozen and unfrozen rocks [7,8].

Experimental investigations on the mechanical properties of geotechnical materials were conducted under freezing conditions [9,10,11,12]. Fener et al. compared test results from different indirect methods and pointed out that empirical predictive results exhibited different behaviors for various rock types [9]. Based on regression analyses, an artificial neural network, adaptive neurofuzzy inference system and physical indices (e.g., point-load strength, block punch index, Schmidt rebound hardness, effective porosity, density) were used to estimate rock compression or tensile strengths [10,13,14]. Many studies indicate that there is no intuitive relation between rocks’ point-load strength index and compression strength [15,16]. Some researchers pointed out that there were significant linear correlations between the point-load strength index and uniaxial compression strength [14]. In cold regions, the widely accepted evaluation of the above rock indices could change at the freezing state. Some decay-function models were proposed to predict the long-term durability of rocks based on their strength deterioration under freezing–thawing or/and drying–wetting [17,18,19,20,21]. Previous investigations indicated that rock compressive and tensile strengths in subzero temperatures were increased as the temperatures were reduced, and the temperature had a more significant effect on the tensile strength than compressive strength [22,23,24,25]. Recently, Zhang et al. systematically investigated the physical and mechanical properties of sandstone subjected to freeze–thaw cycles, chemical erosion, and cyclic loading, and the results implied that the porosity was a key factor attributed to sandstone strength degradation [5,6]. Liu et al. proposed that the combined coal–rock body experienced strain recovery, and this played a loading effect on the failure of the coal [26]. Still, quantitative analyses of the mechanical properties of freezing coal and sandstone are limited, and the effect of unfrozen water content is frequently ignored.

Rock failure is also an important issue in geotechnical engineering. For brittle rock materials in uniaxial and biaxial compression, two types of crack occur: Wing cracks (tensile cracks) and secondary cracks (initiating as shear cracks) [27]. A digital camera through an optical microscope or scanning electron microscope was used to capture the crack patterns under uniaxial loading conditions [28]. Moreover, the applied energy and failure pattern in the Brazilian tensile test was closely related to the inclination angle between the layer plane and loading direction [29]. Tavallali and Vervoort summarized three types of failure patterns of anisotropic rocks in Brazilian tensile tests: Layer activation, central fracture, and noncentral fracture [29]. Thereafter, two failure patterns, central multiple and central/noncentral plus layer, were added by Basu et al. [13]. However, few researchers have paid attention to the failure patterns of rocks under freezing states. 

To better understand the effects of temperature on the mechanical and fracture characteristics of water-bearing sandstone and coal, Brazilian tensile tests, and uniaxial and triaxial compressive tests were performed under various temperature conditions. A predicted relationship between strength and temperature was proposed. Their failure patterns in the tensile and compressive tests considering freezing states were analyzed. 

## 2. Experimental Methodology

### 2.1. Materials and Sample Preparation

The used sandstone and coal samples were taken from the Qinshui basin, Shanxi (37°15′N and 112°45′E). They were obtained using the water-drilling method and then cut to cylinders with a 5.0 cm diameter and 10.0 cm height. All specimens were cut perpendicular to the bedding planes to avoid breakage during preparation. The ratio of the sample length to diameter was 2 for both the uniaxial-compression and triaxial-compression tests, and 0.5 for the Brazilian tensile tests. The samples were saturated with pure water using a vacuum extracting apparatus for 4 h, and then soaked in water for another 4 h. The above procedures were in accordance with American code ASTM D4543-08 [30].

The water content was calculated with the following formula:(1)$$w_{m}=\frac{m_{w}-m_{d}}{m_{d}}\times 100\%$$
where *w*_m_ is the mass water content, and *m*_w_ and *m*_d_ are the weights of the specimens before and after drying, respectively. The basic physical properties of the samples are summarized in Table 1.

### 2.2. Experimental Method

In this study, an electronic universal testing machine CSS-1120 (Sinotest Equipment Co., ltd., Changchun, China) with a maximum axial load of 200 kN was employed to conduct the uniaxial-compression and Brazilian tensile tests. In addition, a set of splitting molds was used in the Brazilian tensile tests. The low-temperature triaxial tests were performed in an improved MTS-810 material test machine (MTS Systems Corporation, Eden Prairie, MN, USA) that can measure the volumetric strain of the specimens. Its maximum axial load was 250 kN, axial displacements were ± 75 mm, and the confining pressure ranged from 0.3 to 35 MPa. Other devices used include a refrigerator with a high-precision automatic temperature-controlling system (temperature range from −30 to 30 °C and resolution of ± 0.5 °C), an electronic scale with a resolution of ± 0.01 g, a Vernier caliper, and a vacuum extracting apparatus. Alcohol was used as a cooling medium in the temperature-control system; the schematic of temperature-control systems was introduced by Zhou et al. [4,31,32,33].

Confining pressures were set as 0, 2, 6, and 8 MPa at −15 °C for the triaxial-compression tests, and temperatures of 20, −4, −10, and −15 °C were selected for the uniaxial-compression and Brazilian splitting tests. The axial loading rate was 0.5 MPa/s for all the tests and the confining loading rate was 0.05 MPa/s for the triaxial-compression tests. Test procedures followed American standard ASTM D7012-10 [34].

## 3. Results and Discussion

### 3.1. Brazilian Tensile Strength 

Typical stress–strain curves of saturated sandstone and coal samples at 20, −4, −10, and −15 °C, which were obtained from the Brazilian tensile tests, are shown in Figure 1, where *W* is the radial displacement of the cylindrical specimen, *D* is the radius, and *W/D* is the radial strain.

The tensile stresses *σ_t_* (Pa) of the rock samples were calculated by the following formula:(2)$$\sigma_{t}=\frac{2F}{\pi Dt}$$
where *F* is the maximum load (N), *D* is the radius (m), and *t* is the thickness of the disc specimen (m).

The Brazilian tensile strength (BTS) is defined by the maximum tensile stress of the stress–strain curves. As shown in Figure 1, the BTS of the saturated sandstone and coal specimens at 20 °C was significantly lower than at subzero temperatures, but their tensile strength slightly changed at subzero temperatures. This indicates that temperature played an important role during the phase-change process of the water. Tensile strength can be expressed as a form of cohesion, and the ice content plays an important role in increasing this cohesion. The water-phase transition in rocks increases the bonding strength between particles. Due to this cementation effect of ice crystals, the tensile strength of freezing rocks was higher than that of unfrozen rock samples. 

Previous researchers proved that unfrozen water still existed in pore media at negative temperatures [35]. Peng stated that unfrozen water content in loess dramatically decreased when the temperature changed from 0 to −5 °C, and the tensile strength of saturated loess significantly increased within this range [36]. He also pointed out that unfrozen water content in pores changed slightly when the temperature was under −5 °C, and the tensile strength was almost constant. Similarly, sandstone and coal are also two porous materials in loess. Then, sandstone and coal tensile strength slightly increased when the temperature decreased from −4 to −15 °C, which could be attributed to the stable unfrozen water content within this range. 

Based on the critical rock mechanics and unfrozen water changes, a tensile-strength prediction is established in this paper. First, the modified effective cementation pressure of ice crystals $\sigma_{t}$ is proposed to describe the tensile-strength-increasing phenomenon, as follows:(3)$$\sigma_{t}=\sigma_{t0}+\sigma_{t}^{*}$$
where $\sigma_{t0}$ is the tensile strength of specimens under unfrozen situations, and $\sigma_{t}^{*}$ is a function of tensile strength, which describes the cementation effect of ice crystals related to ice content. According to the experiment results, $\sigma_{t}^{*}$ can be assumed as the following expression:(4)$$\sigma_{t}^{*}=\frac{e^{\alpha \left(T-T_{f}\right)/T_{0}}}{e^{\alpha \left(T-T_{f}\right)/T_{0}}+1}\sigma_{t-res}$$
where the parameter α is a constant related to the change of ice crystal content, through known data sets of $\sigma_{t}^{*}$ and $\sigma_{t-res}$, α can be obtained by fitting with least-squares [37]; $T_{0}$ is equal to 1 °C, and $\left(T-T_{f}\right)/T_{0}$ is a dimensionless value; $\sigma_{t-res}$ is the increment of tensile strength when water-content changes from a saturated to a residual state. 

As shown in Figure 2, a good correlation between fitting curves based on Equation (3) and the experiment results, was identified (correlation coefficient $R^{2}>0.94$). The optimum parameters of Equation (3), the $\sigma_{t0}$, α, and $\sigma_{t-res}$ of sandstone and coal samples, are presented in Table 2. The fitted results indicated that parameters $\sigma_{t0}$, α, and $\sigma_{t-res}$ were correlated to the geomaterial properties. The transition zones of fitting curves within 0 ~ −5 °C were with respect to the relationship between the unfrozen water and the temperature. The predicted tensile strength remained constant after a certain temperature threshold, which was attributed to stable unfrozen water content.

Compared to the coal samples, a larger decrease of sandstone tensile strength can be seen in Figure 2. There were two main reasons for this difference: (1) Sandstone and coal minerals are different and, thus, their temperature sensitivities are different; (2) coal and sandstone pore-surface areas are different. Their unfrozen water content and tensile strength at the same temperature changed with their physical properties.

### 3.2. Uniaxial-Compression Strength

Uniaxial-compression tests were performed at four temperatures: 20, −4, −10, and −15 °C. The obtained stress–strain curves from the uniaxial-compression tests are shown in Figure 3. Uniaxial-compression strengths (UCSs) at corresponding temperatures were determined by the maximum stresses in the stress–strain curves.

As shown in Figure 3, the uniaxial-compression strength (UCS) increased with the decrease in temperature. The UCS of sandstone was 48.67 MPa at 20 °C, and increased to 56.45 MPa at −15 °C with a 16.0% increment; that of the coal sample was 4.86 MPa at 20 °C, and increased to 5.22 MPa at −15 °C with a 7.4% increment. The UCS increased slightly when the temperature decreased from −4 to −15 °C. As mentioned previously, the water-to-ice phase transition in rock samples increases the bond effect between rock particles. It was verified that the cohesion effect of ice crystals varies with the unfrozen water content of the freezing soils [3,37]. Attributed to the bonding effect of ice crystals, the UCS was higher for rock samples with a higher ice content, and the UCS increase was not significant under −4 °C because of the constant frozen water content. With the temperature decrease, the UCS of the rocks increased and the critical compressive strain decreased [38]. In Figure 3, we can see that the specimens became brittle with the decrease of temperature. The influence of temperature on the UCS and the critical compressive strain was similar to a previous study on sandy mudstone [39].

Similar to tensile strength, the following expressions were adopted to describe the UCS of freezing coal and sandstone.
(5)$$\sigma_{c}=\sigma_{c0}+\sigma_{c}^{*}$$
where $\sigma_{c0}$ is the uniaxial-compression strength of specimens under unfrozen situations and $\sigma_{c}^{*}$ is a function of uniaxial-compression strength, which describes the cementation effect of ice crystals with the change of ice content. $\sigma_{c}^{*}$ is assumed as Equation (6) based on experimental results and previous studies:(6)$$\sigma_{c}^{*}=\frac{e^{\beta \left(T-T_{f}\right)/T_{0}}}{e^{\beta \left(T-T_{f}\right)/T_{0}}+1}\sigma_{c-res}$$
where the parameter β is related to the phase transition process, which reflects the changing rate of ice crystal content, and through known data sets of $\sigma_{c}^{*}$ and $\sigma_{c-res}$, β can be obtained by fitting with least-squares [37]; and $\sigma_{c-\mathrm{res}}$ is the increment between compression strengths at saturated and residual water contents.

As shown in Figure 4, the fitting lines based on Equation (6) are close to the experiment results. The optimum parameters of Equation (6), $\sigma_{c0}$, β, and $\sigma_{c-res}$ for sandstone and coal samples, are presented in Table 3. The correlation coefficients $R^{2}$ between the fitting lines and test data were higher than 0.85. The comparison between Figure 2 and Figure 4 show that the changes of ultimate stresses (tensile and compression strengths) were not significant from −15 to −4 °C, but they obviously changed during the phase-change process. Similar to the tensile strengths, the unfrozen water contents of coal and sandstone at the same temperature changed with their physical properties, subsequently, the compressive strengths also changed.

An established criterion was used to describe the above yielding behavior [40]. For results obtained at the same strain rate, this criterion was expressed by using Nadai’s concept of octahedral shear stress $\tau_{0}$ [41]:(7)$$\tau_{0}+\mu \left(T\right)p=\mathrm{constant}$$
where μ(T) is a parameter related to temperature and p is hydrostatic stress.

The ratio of yield stress in compression $\sigma_{c}$ to yield stress in tension $\sigma_{t}$, calculated from Equation (7) for obtained results at the same temperature and strain rate, is given by:(8)$$\frac{\left|\sigma_{c}\right|}{\sigma_{t}}=\frac{\sqrt{2}+\mu \left(T\right)}{\sqrt{2}-\mu \left(T\right)}$$

The ratios of the UCS to BTS for sandstone and coal samples are shown in Figure 5. They were approximately 6.35 and 2.55 for the sandstone and coal sample, respectively. At the same time, the strength ratios of rock specimens quickly changed at temperatures ranging from 20 to −4 °C, and the ratios remained almost unchanged when the temperature varied at subzero. For example, the strength ratio for sandstone was 7.12 at 20 °C, and it decreased to 6.07 and 6.31 at −4 and −15 °C. For coal samples, the strength ratios were 2.72 at 20 °C and 2.51 at −4 °C, and the ratio at −15 °C was almost as unchanged as that at −4 °C.

Figure 6 illustrates that parameters μ(T) varied with temperature. For sandstone, the  μ(T) value was 1.07 at 20 °C, and decreased to 1.02 when temperatures were below −4 °C. For coal samples, the μ(T) values for sandstone were 0.65 at 20 °C, and 0.61 at negative temperatures. The different values of μ(T) between the sandstone and coal samples indicated that the influences of temperature on the UCS to BTS ratios were different.

### 3.3. Triaxial-Compression Strength

In the triaxial-compression tests on freezing saturated coal specimens at −15 °C, four confining pressures (i.e., 0, 2, 4, and 6 MPa) were chosen. The stress circles, on the basis of the Mohr–Coulomb theory, are presented in Figure 7. The principle stress line at a failure state was obtained on the basis of the stress path, e.g., take $q=\frac{\sigma_{1}-\sigma_{3}}{2}$ as the vertical axis and $p=\frac{\sigma_{1}+\sigma_{3}}{2}$ as the horizontal axis. The slope of the straight line, obtained by connecting the vertex of stress circles, is tanα. The intercept of the vertical axis and this straight line is d. Normally, the internal friction angle ϕ and the cohesion c are expressed as follows:(9)$$\phi =\sin^{-1}(\tan\alpha )$$
(10)$$c=\frac{d}{\cos\phi }$$

Slope tanα=0.569 and intercept d=1.289 MPa were obtained from Figure 7, and the internal friction angle and cohesion were calculated as ϕ=34.72°, c=1.568 MPa.

The Drucker–Prager (DP) yield criterion is a stress-dependent model for determining the failure of normal plastic materials. In the *p–q* space, the yielding function of the DP model, which is dependent on the internal friction angle and cohesion, is expressed as:(11)$$F\left(\phi^{\prime },c^{\prime }\right)=\phi^{\prime }I_{1}-\sqrt{J_{2}}+c^{\prime }=0$$
where $I_{1}$ and $J_{2}$ are the first invariant principal stress and the second invariant deviatoric stress, respectively. Given by Equations (12) and (13), $\phi^{\prime }$ and $c^{\prime }$ are related to the internal friction angle ϕ and the cohesion c.
(12)$$\phi^{\prime }=\frac{\sin\phi }{\sqrt{3}(3+\sin^{2}\phi {)}^{1/2}}=10.31{}^{\circ}$$
(13)$$c^{\prime }=\frac{\sqrt{3}c\cos\phi }{{\left(3+\sin^{2}\phi \right)}^{1/2}}=1.225\;\mathrm{MPa}$$

The deviatoric stress tensor $s_{ij}$ is calculated by $\sigma_{ij}-p\delta_{ij}$, where $\delta_{ij}$ is the Kronecker function. In the generalized *p–q* space, stresses *p* and *q* are expressed by:(14)$$p=\frac{1}{3}\sigma_{ii}=\frac{1}{3}I_{1}$$
(15)$$q=\sqrt{3J_{2}}=\sqrt{\frac{3}{2}s_{ij}s_{ij}}$$
where *p* is confining pressure (i, j=1, 2,3), $\sigma_{ii}=\sigma_{1}+\sigma_{2}+\sigma_{3}$ ($\sigma_{1}\;$and $\sigma_{3}$ are the major and minor principal stresses, respectively).

By substituting Equations (12)–(15) into Equation (11), the following is obtained:(16)q=0.937p+2.121

As shown in Figure 8, the test data of critical *p* and *q* all fell outside of the DP yielding surface, which indicated that actual yielding stresses of freezing saturated coals were higher than normal plastic materials.

### 3.4. Failure Modes

Normally, the failure modes of rocks are explained by inside damage evolution [13]. There are four types of failure modes for cylindrical rocks in Brazilian tensile tests [13,42]. The layer activation (1) is where fractures are parallel to the joint plane. The central fracture (2) is where the fractures are roughly parallel to the loading direction and generally located in the central part of samples. The central part is defined as the range within 10% of the diameter around the axis. The noncentral fracture (3) is where fractures are located outside the central part, and failure planes are not parallel to the loading direction. It is explained by cracks not propagating along the maximum tensile stress, and microcracks dominate this type of fracture. Finally, the central plus layer activation (4) is where layer activation is found in combination with central or noncentral fractures.

The appearance of the failure modes of coal and sandstone is presented in Figure 9. In the case of sandstone, failure modes were central (S-13, S-23) and central multiple (S-33). The failure modes of coal specimens included central (C-31), noncentral (C-22), central multiple (C-12, C-32) and central plus layer activation (C-11, C-21). The sandstone samples failed in the central and central multiple modes, which resulted from horizontal tensile stresses, whereas compression-induced tensile stresses in the coal specimens were partly released along the joints.

Various specimen failure patterns in compression tests are shown in Figure 10. The failure modes were categorized into three types (i.e., axial splitting, shearing along a single plane, and multiple fracturing) [13]. As shown in Figure 10, specimens C8, Y7, and Y4 failed at shearing along a single plane, specimen C12 failed at axial splitting, and multiple fracturing was the dominant failure mode of specimens Y10 and Y13.

Rock failure patterns are closely related to damage evolution, i.e., microcrack generation, propagation, and closure [28,43,44,45]. The sections of unloaded and failed rock specimens in compression tests were extracted [13]. The selected sections of the failed specimens located at the long side parallel to the axis, and were perpendicular to the intersecting line of the failure plane and bottom. Then, it was convenient to track microcrack directions.

During compression, pre-existing microcracks tend to a suitable dimension and orientation with respect to the direction and magnitude of maximum principal stress. Critical compression stress is termed as crack-closure stress. The cracks propagated from the tips when compression- induced tensile stresses exceeded the tensile strength at the pre-existing cracks, named wing cracks, which were parallel to the direction of the maximum principal stress [45]. Previous studies revealed that the dominant failure mode of rock-specimen changes from axial splitting to shearing along a single plane, and to multiple fracturing with the increase of compression stresses [13]. The specimen failed in an axial-splitting pattern when the wing cracks propagated. When the specimen released strain energy in the form of shear fracture, the propagation of adjacent wing cracks along the maximum principal stress was constrained.

## 4. Conclusions

Laboratory tests, such as Brazilian tensile tests, uniaxial tests, and triaxial-compression tests, were conducted to investigate the mechanical behaviors of sandstone and coal at different temperatures. Moreover, their failure patterns were obtained and analyzed. The main conclusions can be summarized as follows:

(1) The tensile and compression strengths of sandstone and coal increased remarkably when temperatures changed from 20 to −4 °C, but they were almost constant as temperature decreased to under the subzero temperature range (i.e., from −4 to −15 °C).

(2) By combining the experimental results, a mathematical model was proposed based on the phase-transition theory, which could be used to predict the relationship between the tensile/compression strength and the temperature. A good correlation between the predictions and measurements was identified.

(3) Four types of failure patterns (i.e., layer activation, central fracture, noncentral fracture, and central plus layer activation) were obtained in the Brazilian tensile tests. Three failure modes (i.e., axial splitting, shearing along a single plane, multiple fracturing) were observed in the compression tests.

(4) By analyzing the sections of unloaded and failed rock specimens in the compression tests, rock damage evolution was divided into four stages: Crack closure (closing of cracks), fracture initiation (linear elastic deformation), critical energy release (stable/unstable fracture propagation), and rupture (coalescence of cracks).

For the practice of the freezing excavation method in coal mining, this paper is significant to understand the mechanical behaviors of sandstone and coal at positive and subzero temperatures.

## Figures and Tables

**Figure 1 materials-12-00992-f001:**
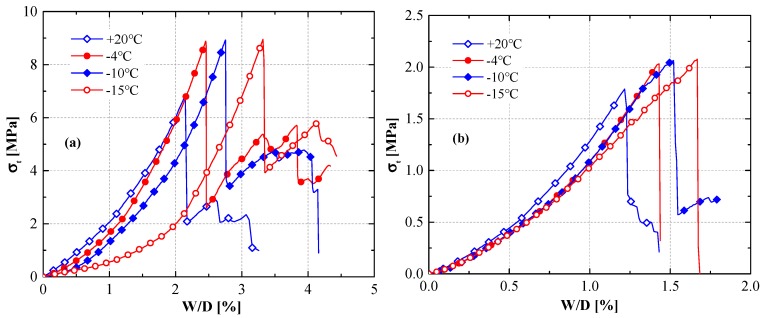
Stress–strain curves in indirect tensile tests under different temperatures: (**a**) sandstone, (**b**) coal samples.

**Figure 2 materials-12-00992-f002:**
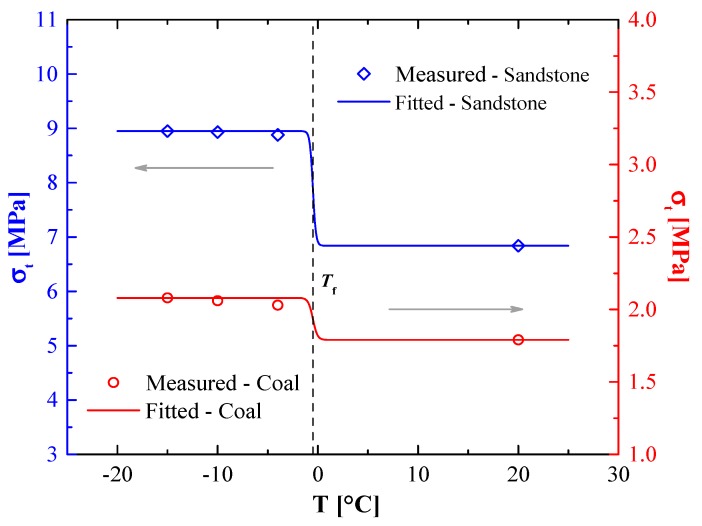
Tensile strengths of sandstones and coal samples as a temperature function.

**Figure 3 materials-12-00992-f003:**
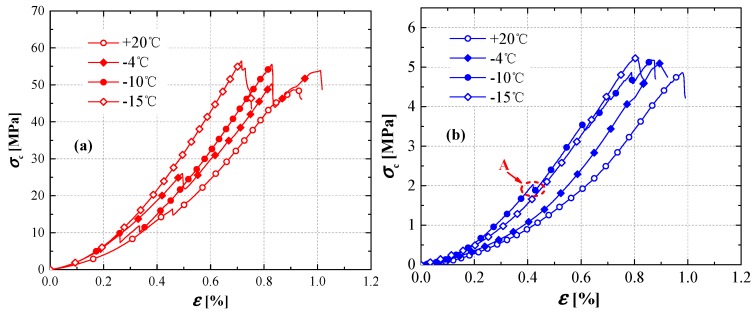
Stress–strain curves from uniaxial-compression tests: (**a**) sandstone; (**b**) coal samples.

**Figure 4 materials-12-00992-f004:**
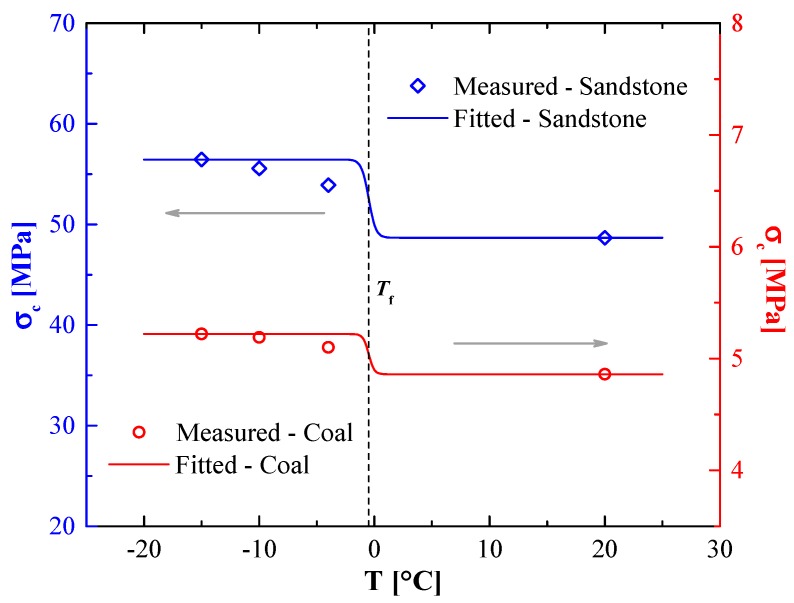
Uniaxial-compression strengths of sandstones and coal samples as a function of temperature.

**Figure 5 materials-12-00992-f005:**
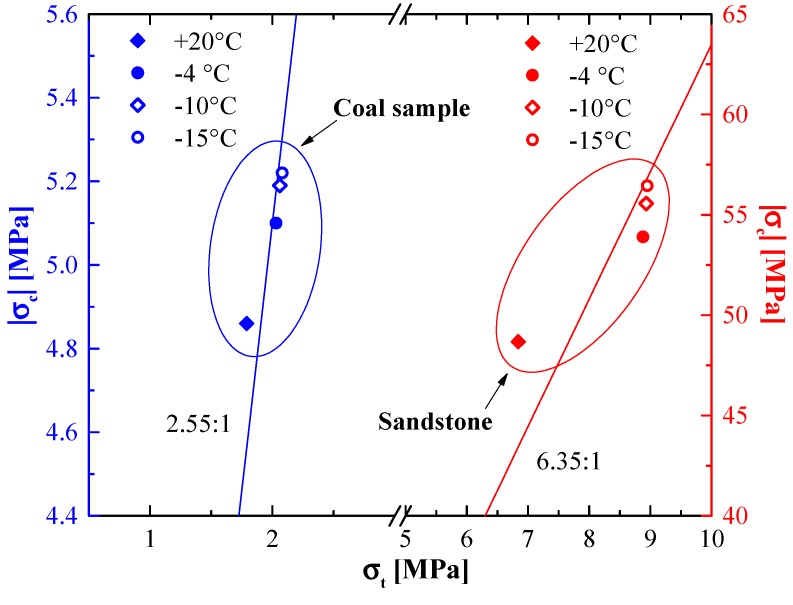
Plot of yield stress in compression versus ultimate tensile stress for performed tests at various temperatures.

**Figure 6 materials-12-00992-f006:**
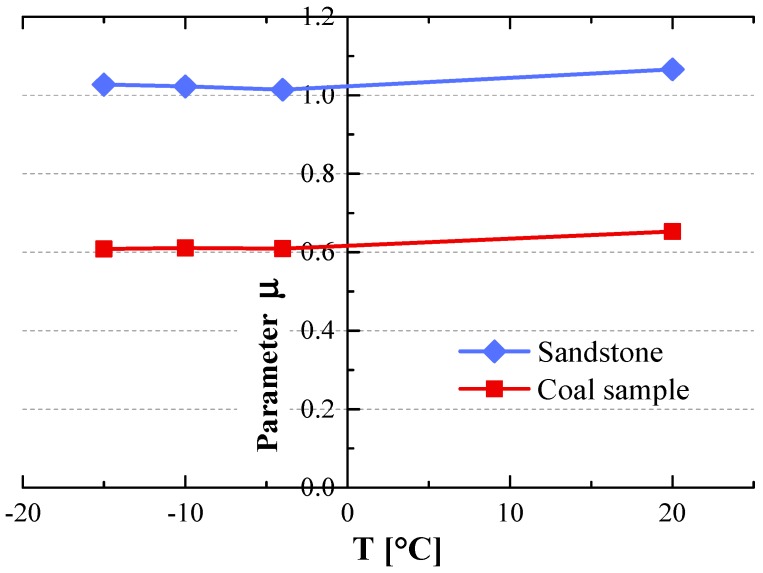
Relationship of parameter μ(T) and temperature.

**Figure 7 materials-12-00992-f007:**
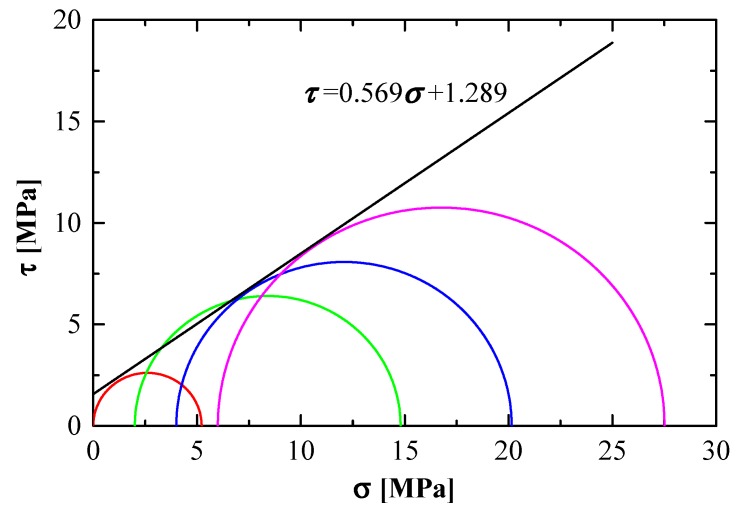
Relationship between mole circle and strength envelope (at −15 °C).

**Figure 8 materials-12-00992-f008:**
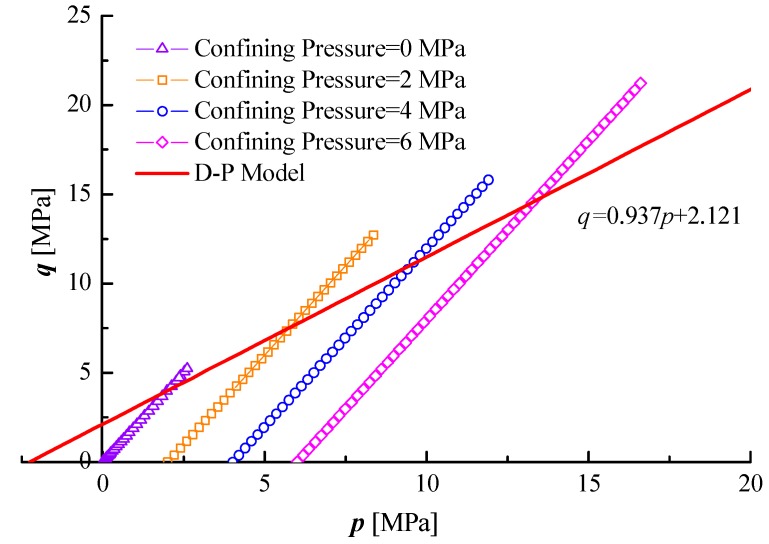
Critical strength test results for freezing coal specimens.

**Figure 9 materials-12-00992-f009:**
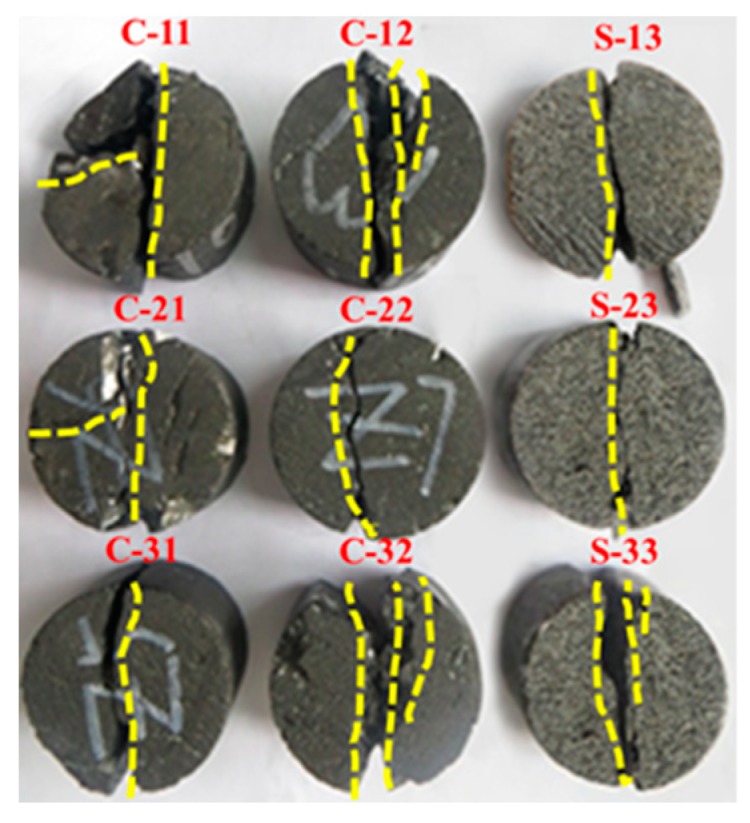
Failure modes observed in coal and sandstone specimens under the Brazilian test.

**Figure 10 materials-12-00992-f010:**
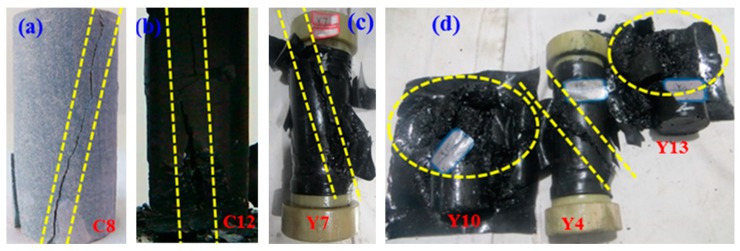
Failure modes observed in coal and sandstone specimens under the compression test.

**Table 1 materials-12-00992-t001:** Sample physical parameters.

Rock Specimen	Density (g·cm^−3^)			Water Content (%)			Effective Porosity (%)
	Dry	Natural	Saturated	Dry	Natural	Saturated	
Sandstone	2.6589	2.6810	2.7273	0	0.8312	2.5725	6.8368
Coal sample	1.4264	1.4414	1.4766	0	1.0539	3.5253	5.0283

**Table 2 materials-12-00992-t002:** Theory parameters (Equation (3)) for tensile tests.

Rock Specimen	Parameter				
	$\sigma_{t0}\;(\mathrm{MPa})$	α	$T_{f}\;({}^{\circ}C)$	$\sigma_{t-res}\;(\mathrm{MPa})$	*R* ^2^
Sandstone	6.84	−6.680	−0.5	2.11	0.9984
Coal sample	1.79	−4.417	−0.5	0.29	0.9469

**Table 3 materials-12-00992-t003:** Theory parameters (Equation (6)) for uniaxial-compression experiments.

Rock Specimen	Parameter				
	$\sigma_{c0}\;(\mathrm{MPa})$	β	$T_{f}\;({}^{\circ}C)$	$\sigma_{c-res}\;(\mathrm{MPa})$	*R* ^2^
Sandstone	48.67	−3.256	−0.5	7.78	0.8995
Coal sample	4.86	−4.239	−0.5	0.36	0.8585

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
