# Peer review of "Experimental Study on Mechanical Characteristics and Fracture Patterns of Unfrozen/Freezing Saturated Coal and Sandstone"

_materials, 2019, doi:10.3390/ma12060992_

Reviewer 1 Report

The Authors performed the interesting study which is worth being published. There are very few investigations on frozen coal. However I have some general and detailed remarks.

The main general remark is excess of cited papers which are not related to rocks (they concern soils as loess or sand) or to the other rock types as granite. The soil structure as the cohessive materials and aluminosilicates reaction with water leads to completely different behaviour with water of them. Granite - as the igneous rock with another type of bonds of minerals also cannot be a good example of wet rock behaviour under mechanical tests. Of course it is a solid rock, but I wouldn't use too many such examples.

So I suggest not to quote on them and withdraw several of them ca. 15 positions (e.g. Gullu et all about CBR test - for rocks?!).

The second general remark is that I don't know how did you keep the low temperature (the frost state) during the tests? Comment it, please.

The detailed remarks are as follows:

page 4, line 131 - replace the word "was" with "changed"

page 4, line 146 - you don't need space before the word "water"

page 4, formula 4 (page 6, formula 6) - explain, please, what does it mean "T" and "To". Two variables? I coudn't check the values.

page 4, line 151 - correlated "to"

page 5, fig. 3 - use the same signs and colours for the same temperature lines

page 6, line 197 - I would prefer the expression "ultimate stress" than "yield stress" here

page 7, line 215 - 6.35 and 2..5 are extremely low proportions. Comment it, please

page 7, line 215 - the "strength" ratios

page 7, line 216 - of "rock" specimens

page 7, line 217 - the ratio of "strengths for" sandstone

page 7, line 218 - For coal "samples the strength" ratios

page 8, figure 5 - indicate the different temperatures. Maybe use different sign shapes?

page 9, line 241 - Φ = 0.6056 "rad" or 34.70

page 9, line 251 and 252 - what units? I've got the different numbers...

page 10, figure 8 - the chart is unclear. What does the red line show? Why not to plot the p-q line for the experiment. Where is the full p-q chart for your experiment

page 11, line 300 - the type error - "single", lack of "n"

page 12, lines 333-335 - font size!

page 13, line 350 - ASTM - capital letters

page 13, line 359 - hyphen separation

page 13, line 369 - one hyphen and separation

page 13, line 377-378 - small letters

page 14, line 422-423 - small letters

Author Response

Reviewer 1:

The Authors performed the interesting study which is worth being published. There are very few investigations on frozen coal. However I have some general and detailed remarks.

The main general remark is excess of cited papers which are not related to rocks (they concern soils as loess or sand) or to the other rock types as granite. The soil structure as the cohessive materials and aluminosilicates reaction with water leads to completely different behaviour with water of them. Granite - as the igneous rock with another type of bonds of minerals also cannot be a good example of wet rock behaviour under mechanical tests. Of course it is a solid rock, but I wouldn't use too many such examples. So I suggest not to quote on them and withdraw several of them ca. 15 positions (e.g. Gullu et all about CBR test - for rocks?!).

Response: these have been revised based on the comment.

Firstly, the reviews on literatures such as Gullu et al. (2016), Chai et al. (2018), Kozlowski et al. (2018), Li et al. (2009), Takarli et al. (2003), Tang et al. (2010), Xu et al. (2011), Xu et al. (2016b), Yang et al. (2010b), Yesiloglu-Gultekin (2013) were deleted.

Chai, M., Zhang, J., Zhang, H., Mu, Y., Sun, G. and Yin, Z., 2018. A method for calculating unfrozen water content of silty clay with consideration of freezing point. Applied Clay Science, 161: 474-481.

Güllü, H. and Fedakar, H.İ., 2016. Use of factorial experimental approach and effect size on the CBR testing results for the usable dosages of wastewater sludge ash with coarse-grained material. European Journal of Environmental & Civil Engineering: 1-22.

Kozlowski, T., 2003. A comprehensive method of determining the soil unfrozen water curves: 1. Application of the term of convolution. Cold Regions Science & Technology, 36(1): 71-79.

Li, S., Lai, Y., Zhang, S. and Liu, D., 2009. An improved statistical damage constitutive model for warm frozen clay based on Mohr–Coulomb criterion. Cold Regions Science & Technology, 57(2–3): 154-159.

Takarli, M., Prince, W. and Siddique, R., 2008. Damage in granite under heating/cooling cycles and water freeze–thaw condition. International Journal of Rock Mechanics & Mining Sciences, 45(7): 1164-1175.

Tang, M., Wang, Z., Sun, Y. and Jinhong, B.A., 2010. EXPERIMENTAL STUDY OF MECHANICAL PROPERTIES OF GRANITE UNDER LOW TEMPERATURE. Chinese Journal of Rock Mechanics & Engineering, 29(4): 787-794.

Xu, X., Lai, Y., Dong, Y. and Qi, J., 2011. Laboratory investigation on strength and deformation characteristics of ice-saturated frozen sandy soil. Cold Regions Science & Technology, 69(1): 98-104.

Xu, X., Wang, Y., Bai, R., Zhang, H. and Hu, K., 2016b. Effects of sodium sulfate content on mechanical behavior of frozen silty sand considering concentration of saline solution. Results in Physics, 6(C): 1000-1007.

Yang, Y., Lai, Y. and Li, J., 2010b. Laboratory investigation on the strength characteristic of frozen sand considering effect of confining pressure. Cold Regions Science & Technology, 60(3): 245-250.

Yesiloglu-Gultekin, N., Gokceoglu, C. and Sezer, E.A., 2013. Prediction of uniaxial compressive strength of granitic rocks by various nonlinear tools and comparison of their performances. International Journal of Rock Mechanics & Mining Sciences, 62(9): 113-122.

Secondly, Bauwens-Crowet (1972) was cited for introducing the yielding criterion (Eq. 7). Xu et al. (2016a), Xu et al. (2017) were cited to introduce the freezing excavation method. Li et al. (2013) was cited to introduce the digital camera method. Liao et al. (2016) and Peng et al. (1998) presented the cohesion effect of ice crystals and frozen temperatures in porous media. Zhou et al. (2016a, 2016b, 2018a, 2018b) introduced the set-up of temperature controlling system. Therefore, the above literatures were kept in the revised manuscript.

Bauwens-Crowet, C., Bauwens, J.C. and Homès, G., 1972. The temperature dependence of yield of polycarbonate in uniaxial compression and tensile tests. Journal of Materials Science, 7(2): 176-183.

Li, L., Lee, P.K.K., Tsui, Y., Tham, L.G. and Tang, C.A., 2003. Failure Process of Granite. International Journal of Geomechanics, 3(1): 84-98.

Liao, M., Lai, Y. and Wang, C., 2016. A strength criterion for frozen sodium sulfate saline soil. Canadian Geotechnical Journal, 53(7): 1176-1185.

Peng, W., 1998. Tensile strength of frozen loess varying with strain rate and temperature. Chinese Jounal of Geotechnical Engineering, 20(3).

Xu, X., Wang, Y., Bai, R., Fan, C. and Hua, S., 2016a. Comparative studies on mechanical behavior of frozen natural saline silty sand and frozen desalted silty sand. Cold Regions Science & Technology, 132: 81-88.

Xu, X., Wang, Y., Yin, Z. and Zhang, H., 2017. Effect of temperature and strain rate on mechanical characteristics and constitutive model of frozen Helin loess. Cold Regions Science & Technology, 136: 44-51.

Zhou, Z.W., Ma, W., Zhang, S.J., Du, H.M., Mu, Y.H., Li, G.Y., 2016a. Multiaxial creep of frozen loess. Mech. Mater. 95, 172–191.

Zhou, Z.W., Ma, W., Zhang, S.J., Mu, Y.H., Zhao, S.P., Li, G.Y., 2016b. Yield surface evolution for columnar ice. Results Phys 6, 851–859.

Zhou, Z., Ma, W., Zhang, S., et al., 2018a. Damage evolution and recrystallization enhancement of frozen loess. Int. J. Damage Mech 27, 1131–1155.

Zhou, Z., Ma, W., Zhang, S., Mu, Y., Li, G., 2018b. Effect of freeze-thaw cycles in mechanical behaviors of frozen loess. Cold Reg. Sci. Technol. 146, 9-18.

The second general remark is that I don't know how did you keep the low temperature (the frost state) during the tests? Comment it, please.

Response: these have been revised based on the comment, see lines 96-97.

Zhou et al. (2016a; 2016b; 2018a; 2018b) presented a schematic of temperature control systems for the uniaxial compression tests, Brazilian tensile tests and triaxial tests. The alcohol is used as cooling medium in the temperature control system, and the range value of testing temperature is from 30 ℃ to -30 ℃. The set-up of this test was shown in the attachment.

Zhou, Z.W., Ma, W., Zhang, S.J., Du, H.M., Mu, Y.H., Li, G.Y., 2016a. Multiaxial creep of frozen loess. Mech. Mater. 95, 172–191.

Zhou, Z.W., Ma, W., Zhang, S.J., Mu, Y.H., Zhao, S.P., Li, G.Y., 2016b. Yield surface evolution for columnar ice. Results Phys 6, 851–859.

Zhou, Z., Ma, W., Zhang, S., et al., 2018a. Damage evolution and recrystallization enhancement of frozen loess. Int. J. Damage Mech 27, 1131–1155.

Zhou, Z., Ma, W., Zhang, S., Mu, Y., Li, G., 2018b. Effect of freeze-thaw cycles in mechanical behaviors of frozen loess. Cold Reg. Sci. Technol. 146, 9-18.

The detailed remarks are as follows:

page 4, line 131 - replace the word "was" with "changed"

Response: this have been revised based on the comment, see line 125.

page 4, line 146 - you don't need space before the word "water"

Response: this have been revised based on the comment, see line 141.

page 4, formula 4 (page 6, formula 6) - explain, please, what does it mean "T" and "To". Two variables? I coudn't check the values.

Response: and  is a dimensionless value, these have been emphasized in revised manuscript; see lines 140-141.

page 4, line 151 - correlated "to"

Response: this have been revised based on the comment, see line 146.

page 5, fig. 3 - use the same signs and colours for the same temperature lines

Response: these have been revised based on the comment, see fig. 3.

page 6, line 197 - I would prefer the expression "ultimate stress" than "yield stress" here

Response: these have been revised based on the comment, see lines 191, 216.

page 7, line 215 - 6.35 and 2..5 are extremely low proportions. Comment it, please

Response: Hoek and Brown (1997) stated that the ratio of uniaxial compression strengths to tensile strengths decreased with the uniaxial compression strengths decreasing. Liu et al. (2018) investigated the unconfined compression strengths of pure coal and coal-rock combined body, and the results showed that ultimate compression stresses were under 15 MPa. Wang et al. (2018) pointed out that the strengths of saturated coal samples were 40% of natural samples’ strengths. Guo et al. (2019) reported that the scale characteristics of the pores and fractures in coal will directly affect the calculation of the strength of coals. Then, under the impacts of different porosity and water content, the extremely low ultimate compression stresses and the ratios of UCSs to BTSs of sandstone were probably influenced by the variety of porosity and water content of coals. Similarly, based on previous works, the extremely low proportions of UCSs to BTSs may be attributed to the different porosity and water content of native sandstones (Dyke and Dobereiner, 1991; Al-Harthi et al., 1999; Palchik, 1999; Hawkins A B and McConnell, 1992).

References:

Hoek E, Brown E T. Practical estimates of rock mass strength[J]. International journal of rock mechanics and mining sciences, 1997, 34(8): 1165-1186.

Liu X S, Tan Y L, Ning J G, et al. Mechanical properties and damage constitutive model of coal in coal-rock combined body[J]. International Journal of Rock Mechanics and Mining Sciences, 2018, 110: 140-150.

Guo H, Yuan L, Cheng Y, et al. Experimental investigation on coal pore and fracture characteristics based on fractal theory[J]. Powder Technology, 2019, 346:341-349

Wang W, Wang H, Li D, et al. Strength and failure characteristics of natural and water-saturated coal specimens under static and dynamic loads[J]. Shock and Vibration, 2018.

Dyke C G, Dobereiner L. Evaluating the strength and deformability of sandstones[J]. Quarterly Journal of Engineering Geology, 1991. 24:123-134

Al-Harthi A A, Al-Amri R M, Shehata W M. The porosity and engineering properties of vesicular basalt in Saudi Arabia[J]. Engineering Geology, 1999, 54(3-4): 313-320.

Palchik V. Influence of porosity and elastic modulus on uniaxial compressive strength in soft brittle porous sandstones[J]. Rock Mechanics and Rock Engineering, 1999, 32(4): 303-309.

Hawkins A B, McConnell B J. Sensitivity of sandstone strength and deformability to changes in moisture content[J]. Quarterly Journal of Engineering Geology and Hydrogeology, 1992, 25(2): 115-130.

page 7, line 215 - the "strength" ratios

Response: this have been revised based on the comment, see line 210.

page 7, line 216 - of "rock" specimens

Response: this have been revised based on the comment, see line 210.

page 7, line 217 - the ratio of "strengths for" sandstone

Response: these have been revised based on the comment, see lines 211-212.

page 7, line 218 - For coal "samples the strength" ratios

Response: these have been revised based on the comment, see lines 212-213.

page 8, figure 5 - indicate the different temperatures. Maybe use different sign shapes?

Response: these have been revised based on the comment, see fig. 5.

Figure 5. Plot of the yield stress in compression versus the ultimate tensile stress for tests performed at various temperatures.

page 9, line 241 - Φ = 0.6056 "rad" or 34.70

Response: these have been revised based on the comment, see lines 236, 246.

page 9, line 251 and 252 - what units? I've got the different numbers...

Response: these have been revised based on the comment, see lines 246-247.

page 10, figure 8 – the chart is unclear. What does the red line show? Why not to plot the p-q line for the experiment. Where is the full p-q chart for your experiment

Response: these have been revised based on the comment, see fig. 8.

Figure 8. Critical strength test results for the freezing coal specimens.

page 11, line 300 - the type error - "single", lack of "n"

Response: this have been revised based on the comment, see fig. 11 in the manuscript.

page 12, lines 333-335 - font size!

Response: these have been revised based on the comment, see lines 322-325.

page 13, line 350 - ASTM - capital letters

Response: these have been revised based on the comment, see References.

page 13, line 359 - hyphen separation

Response: these have been revised based on the comment, see References.

page 13, line 369 - one hyphen and separation

Response: these have been revised based on the comment, see References.

page 13, line 377-378 - small letters

Response: these have been revised based on the comment, see References.

page 14, line 422-423 - small letters

Response: these have been revised based on the comment, see References.

Reviewer 2 Report

The paper need a major revision. The comments are as follow:

The writing of the paper is very weak. The problems are too many to mention. An English editor should review the paper and the writing to be improved.

Line 10: What is meant by “during freezing excavation”?

Lines 22-24: check the font size.

There are some new publications on the effect of freeze-thaw cycles on rock properties which are overlooked in this study. A few of them are given below:

Deterioration and strain energy development of sandstones under quasi-static and dynamic loading after freeze-thaw cycles, Cold Regions Science and Technology, 2019, 155:37-46.

Degradation of physical and mechanical properties of sandstone subjected to freeze-thaw cycles and chemical erosion, Cold Regions Science and Technology, 2018, 160: 252-264.

Section 2.2: Please give the size of the samples.

Line 111-115: Check and revise the formatting.

In Equation 4, how constant Alfa is determined? You need to provide a detailed description.

Lines 157-158: Revise the writing.

“3.2. Uniaxial compression strengths” should be: “3.2. Uniaxial compression strength” apply this also to 3.1 and

Equation 5: explain in detail how Beta should be calculated.

Figure 5: Include the temperature for each data point.

2.55 as the ration between compressive and tensile strength seems too low. In general, the ratio should be close to 10. Any explanation for this?

Figure 7: Why two failure lines are drawn?

Lines 309-313: Remove the blue colour.

Author Response

The paper need a major revision. The comments are as follow:

The writing of the paper is very weak. The problems are too many to mention. An English editor should review the paper and the writing to be improved.

Line 10: What is meant by “during freezing excavation”?

Response: these have been revised based on the comment, see lines 10-11, 25.

Lines 22-24: check the font size.

Response: these have been revised based on the comment, see lines 22-24.

There are some new publications on the effect of freeze-thaw cycles on rock properties which are overlooked in this study.

A few of them are given below:

Deterioration and strain energy development of sandstones under quasi-static and dynamic loading after freeze-thaw cycles, Cold Regions Science and Technology, 2019, 155:37-46.

Degradation of physical and mechanical properties of sandstone subjected to freeze-thaw cycles and chemical erosion, Cold Regions Science and Technology, 2018, 160: 252-264.

Response: these have been revised based on the comment, see lines 343-346.

These new literatures were cited in the revised manuscript such as:

Recently, Zhang et al. (2018,2019) systematically investigated the physical and mechanical properties of sandstone subjected to freeze-thaw cycles, chemical erosion and cyclic loading, and results implied that porosity was a key factor attributed to the degradation of strengths of sandstones. Liu et al. (2018) proposed that the coal-rock combined body experienced a strain recovery, and it played a loading effect on the failure of coal.

References:

Zhang J, Deng H, Taheri A, et al. Degradation of physical and mechanical properties of sandstone subjected to freeze-thaw cycles and chemical erosion[J]. Cold Regions Science and Technology, 2018, 155: 37-46.

Zhang J, Deng H, Taheri A, et al. Deterioration and strain energy development of sandstones under quasi-static and dynamic loading after freeze-thaw cycles[J]. Cold Regions Science and Technology, 2019, 160:252-264

Liu X S, Tan Y L, Ning J G, et al. Mechanical properties and damage constitutive model of coal in coal-rock combined body[J]. International Journal of Rock Mechanics and Mining Sciences, 2018, 110: 140-150.

Section 2.2: Please give the size of the samples.

Response: these have been emphasized in revised manuscript; see lines 75-76.

Line 111-115: Check and revise the formatting.

Response: these been revised based on the comment, see lines 105-109.

In Equation 4, how constant Alfa is determined? You need to provide a detailed description.

Response: these have been described based on the comment, see lines 139-140.

Lines 157-158: Revise the writing.

Response: these have been revised based on the comment, see lines 152-153.

“3.2. Uniaxial compression strengths” should be: “3.2. Uniaxial compression strength” apply this also to 3.1 and

Response: these have been revised based on the comment, see lines 104, 158, 225.

Equation 5: explain in detail how Beta should be calculated.

Response: these have been described based on the comment, see line 186.

Figure 5: Include the temperature for each data point.

Response: these have been revised based on the comment, see fig. 5.

2.55 as the ration between compressive and tensile strength seems too low. In general, the ratio should be close to 10. Any explanation for this?

Response: Hoek and Brown (1997) stated that the ratio of uniaxial compression strengths to tensile strengths decreased with the uniaxial compression strengths decreasing. Liu et al. (2018) investigated the unconfined compression strengths of pure coal and coal-rock combined body, and the results showed that ultimate compression stresses were under 15 MPa. Wang et al. (2018) pointed out that the strengths of saturated coal samples were 40% of natural samples’ strengths. Therefore, the results of compression strengths in this paper were within a reasonable range. Guo et al. (2019) reported that the scale characteristics of the pores and fractures in coal will directly affect the calculation of the strength of coals. Then, under the impacts of different porosity and water content, the extremely low ultimate compression stresses and the ratios of UCSs to BTSs were probably influenced by the variety of porosity and water content of coals. Similarly, based on previous works, the extremely low proportions of UCSs to BTSs of sandstone may be attributed to the different porosity and water content of native sandstones (Dyke and Dobereiner, 1991; Al-Harthi et al., 1999; Palchik, 1999; Hawkins A B and McConnell, 1992).

References:

Hoek E, Brown E T. Practical estimates of rock mass strength[J]. International journal of rock mechanics and mining sciences, 1997, 34(8): 1165-1186.

Liu X S, Tan Y L, Ning J G, et al. Mechanical properties and damage constitutive model of coal in coal-rock combined body[J]. International Journal of Rock Mechanics and Mining Sciences, 2018, 110: 140-150.

Guo H, Yuan L, Cheng Y, et al. Experimental investigation on coal pore and fracture characteristics based on fractal theory[J]. Powder Technology, 2019, 346:341-349

Wang W, Wang H, Li D, et al. Strength and failure characteristics of natural and water-saturated coal specimens under static and dynamic loads[J]. Shock and Vibration, 2018.

Dyke C G, Dobereiner L. Evaluating the strength and deformability of sandstones[J]. Quarterly Journal of Engineering Geology, 1991. 24:123-134

Al-Harthi A A, Al-Amri R M, Shehata W M. The porosity and engineering properties of vesicular basalt in Saudi Arabia[J]. Engineering Geology, 1999, 54(3-4): 313-320.

Palchik V. Influence of porosity and elastic modulus on uniaxial compressive strength in soft brittle porous sandstones[J]. Rock Mechanics and Rock Engineering, 1999, 32(4): 303-309.

Hawkins A B, McConnell B J. Sensitivity of sandstone strength and deformability to changes in moisture content[J]. Quarterly Journal of Engineering Geology and Hydrogeology, 1992, 25(2): 115-130.

Figure 7: Why two failure lines are drawn?

Response: after being carefully checked on fig. 7, the gray line is not correlated to the context of this paper. It has been revised based on your useful comment, see fig. 7.

Lines 309-313: Remove the blue colour.

Response: these have been revised based on the comment, see lines 301-306.

Round  2

Reviewer 2 Report

The paper has not been improved in terms of writing. As requested earlier an English editor should revise and improve the paper.

Explanations of how Alfa and Beta are calculated are not clear and sufficient. It is not enough to mention that they are defined based on fitting. The fitting procedure should be explained clearly. If required figures should be provided to make the point clear.

Author Response

Response to reviewers’ comments

Ref: materials-452440

Title: Experimental study on the mechanical characteristic and fracture pattern of unfrozen/freezing saturated coal and sandstone

Journal: Materials

We much appreciate the constructive and insightful suggestions made by the reviewers. We have addressed all of the comments, and list below each comment and our response (in italics).

Corresponding Author: Shuangyang Li ([email protected])

Chong Wang, Shuangyang Li, Tongwei Zhang, Zhemin You

([email protected]; [email protected]; [email protected]; [email protected])

The paper has not been improved in terms of writing. As requested earlier an English editor should revise and improve the paper.

Response: The paper has been improved in terms of writing by MDPI for English editing. The revisions were marked in blue in the manuscript.

Explanations of how Alfa and Beta are calculated are not clear and sufficient. It is not enough to mention that they are defined based on fitting. The fitting procedure should be explained clearly. If required figures should be provided to make the point clear.

Response: Liao et al. (2016) gave the similar formulas as Eqs. (3)-(6) to express the strength of frozen soil. Based on their models, Eqs. (3)-(6) were also selected in our study. A common mathematical method ‘least-squares’ was used to fit the results and the selected formula. These have been revised in manuscript; see lines 139-140, 186-188. The ‘least-squares’ was introduced as follows:

The method of least squares is a standard approach in regression analysis to approximate the solution of over-determined systems, i.e., sets of equations in which there are more equations than unknowns. "Least squares" means that the overall solution minimizes the sum of the squares of the residuals made in the results of every single equation. The most important application is in data fitting. The best fit in the least-squares sense minimizes the sum of squared residuals (a residual being: the difference between an observed value, and the fitted value provided by a model).
